# Evaluation of Body and Udder Temperatures and Mammary Gland Health Status Throughout Lactation in Manchega Dairy Sheep

**DOI:** 10.3390/ani15060773

**Published:** 2025-03-09

**Authors:** Joel Bueso-Ródenas, María Moreno-Manrique, Pilar Gascó, Ramón Arias, Gema Romero, José Ramón Díaz

**Affiliations:** 1Department of Animal Production and Public Health, Faculty of Veterinary Medicine and Experimental Sciences, Catholic University of Valencia San Vicente Mártir, 46001 Valencia, Spain; 2Doctoral School, Catholic University of Valencia San Vicente Mártir, 46001 Valencia, Spain; maria.morenomanrique@gmail.com; 3Departamento de Tecnología Agroalimentaria, Universidad Miguel Hernández (UMH), Ctra. de Beniel km 3.2, 03312 Orihuela, Spaingemaromero@umh.es (G.R.); jr.diaz@umh.es (J.R.D.); 4Centro Regional de Selección y Reproducción Animal (CERSYRA), Instituto Regional de Investigación y Desarrollo Agroalimentario y Forestal de la Junta de Comunidades de Castilla-La Mancha (IRIAF), CERSYRA Valdepeñas, 13300 Ciudad Real, Spain; rarias@jccm.es

**Keywords:** mastitis, dairy sheep, somatic cell count, small ruminants, infrared thermography, ambient temperature, rectal temperature, udder health

## Abstract

This study examined how surface and rectal temperatures in Manchega sheep respond to environmental changes and infection indicators. Surface temperature varied with external weather conditions, while rectal temperature remained stable, suggesting sheep can adjust their physiological response to mild climatic variations without compromising core body temperature. The temperature of the udder varied according to the udder health status. This research helps improve animal health monitoring, which can lead to better disease management in livestock, improving animal welfare and productivity.

## 1. Introduction

Infrared thermography (IRT) has become a valuable and non-invasive tool for assessing health and welfare in domestic animals like pigs and horses to test injuries or stress [1,2]. It has also been used to assess the correspondence between different body temperature measurement methods, such as rectal temperature and surface temperature for different purposes, revealing significant variations across different body regions and environmental conditions [3,4,5]. IRT has also revealed that well-adapted indigenous breeds, such as the Manchega sheep, possess the ability to dissipate excess body heat under hot conditions through increased respiration rates and vasodilation among other adaptive capabilities [5].

IRT technology enables the rapid evaluation of body surface temperature, facilitating the identification of inflammatory processes or infections that alter blood circulation and, consequently, local skin temperature, like in very typical animal disorders such as foot rot or lameness [6,7]. Moreover, IRT can reveal unventilated areas of the nasal cavity with an increase in temperature compared to the contralateral nasal cavity, allowing for presumptive diagnoses of enzootic nasal adenomatosis [8]. In cows, the results about the evaluation of udder health through IRT are discrepant among themselves [9,10,11,12,13,14,15,16,17]. Scott et al. [9] demonstrated that udder quarters exposed to *E. coli* LPS (Lipopolysaccharides) showed a significant temperature increase of about 2.3 °C six hours post-treatment, with this change detectable in the udder’s maximum and average temperatures, but not in unaffected quarters. In a study by Colak et al. [10], a high correlation (r = 0.92) was found between the udder skin surface temperature and the results of the California Mastitis Test (CMT). Other studies by Hovinen et al. [11] and Metzner et al. [12] also confirmed an elevation in the udder surface temperature by IRT after the inoculation of *E. coli* LPS. The first studies by Colak et al. [10] and other more recent studies by Pampariene et al. [13] and Zaninnelli et al. [14] have confirmed a relation between the udder surface temperature and traditional udder health indicators such as SCC and the results of CMT in field conditions. However, Bortolami et al. [15] found that SCC was negatively associated with the udder surface temperature, highlighting that IRT did not have the capability to differentiate mastitis-causing pathogens. Similarly, Porcionato et al. [16] concluded that IRT was not useful for diagnosing mastitis. Also, Byrne et al. [17] did not find a relationship between SCC and udder skin surface temperature.

In small ruminants, such as Manchega sheep, mastitis is commonly caused by infectious bacteria, including various species of *Staphylococcus* spp., which often lead to a high prevalence of subclinical infections [18,19,20]. In this sense, the early detection of mastitis in small ruminants has become a crucial objective in terms of animal health and welfare [21,22], milk quality [23] and safety [24], and farm profitability [25]. Thus, like SCC, electrical conductivity and other methods, also explored on cows, IRT has also been tested as a tool for the early detection of mastitis with scarce success. Martins et al. [26] were able to detect, by IRT, sheep with values of SCC between 250,000 and 500,000 but failed to detect those with values higher than 500,000. Castro-Costa et al. [27], in two experiments, one after the inoculation of *E. coli* LPS and another in field conditions, were unable to discriminate between healthy and infected (subclinically or clinically) udder halves in dairy ewes. A recent study by Marnet et al. [28] concluded that IRT in goats, even when dealing with very high levels of inflammation typically associated with infections from both minor and major pathogens, is not effective for predicting mammary infections.

In Manchega sheep, the correlation of temperatures between different body regions under varying environmental temperature conditions and different stages of lactation has not been thoroughly studied. Specific data on SCC levels, mastitis rates, and udder temperatures remain limited for this breed, highlighting the need for targeted research to improve diagnostic accuracy. The present study specifically aims to correlate udder surface temperature with overall body temperature, environmental temperature, and variables related to udder health status in sheep, focusing on somatic cell count (SCC) and milk microbiological culture.

## 2. Materials and Methods

The Institutional Animal Care and Use Committee (IACUC) of Miguel Hernández University of Elche approved the referenced Animal Use (UMH.DTA.JDS.001.09, date of approval: 15 July 2009).

### 2.1. Animal Facilities and Management

The research took place at the Manchega Ewe National Herd’s premises in Valdepeñas, Spain, under the oversight of the National Association of Select Manchega Sheep Breeders. Animals were housed in barns in which the floor was covered with cereal straw and had access to sunny yards. The stocking density was approximately 2 square meters per animal in the covered areas. The experiment took place between January and July. The temperature inside the facilities ranged from a minimum of 9 degrees to a maximum of 30 degrees, which are typical temperatures for this geographical location and time of year. Every month, in a single control, thermographic images were taken, and mammary gland health statuses were checked (see Section 2.2 and Section 2.3). The ewes were provided with a consistent diet consisting of a blend of cereal grains and alfalfa hay, fed twice daily, with unrestricted access to water. The facility was equipped with a high-line milking system from DeLaval (Tumba, Sweden), set up in a 2 × 18 Casse type milking parlor. Milking procedures occurred routinely twice per day, at 08:00 and 17:00, in a room with a milking parlor with a capacity for 36 sheep.

A preliminary phase lasting 15 days was conducted to identify the animals suitable for this study and to establish their baseline conditions. Initially, a group of 150 ewes with a similar lambing date (4 ± 1 weeks postpartum) was assessed, during which milk yield and mammary gland health were recorded. Based on these results, 108 ewes producing more than 0.85 kg of milk and free of clinical mastitis were chosen, including 80 multiparous and 28 primiparous individuals. These 108 animals were randomly allocated in three similar pens with a capacity for 36 sheep each and were maintained in the same conditions throughout the experiment.

### 2.2. Temperature Measurement and Thermographic Image Description

Once a month, before the morning milking, the animals were led to the milking room where rectal temperatures and thermographic images were captured using a thermographic camera, Flyr ThermaCAM TM SC 660 from FLIR SYSTEMS (Danderyd, Sweden). Ambient temperature (°C) and relative humidity (%) were evaluated every 15 min at the place where the measurements were taken by using a Testo 174 h datalogger (Barcelona, Spain). A digital thermometer Vet 12 (Hauptner and Herberholz, Solingen, Germany) was inserted 3 cm into the rectum to record rectal temperature. The camera was used to record images from three areas: two from the animal’s caudal part to capture the posterior view of the right mammary gland and the left mammary gland and also the perianal area. One additional image was taken from the front of the animal to capture the right lacrimal area. The camera was positioned on a tripod to ensure stability, one meter away from the subject. The camera’s accuracy is within +/−1 °C or 1% of the reading in a restricted range, with a resolution of 640 × 480 pixels, a thermal sensitivity of 30 mK at 30 °C, and a spectral range of 7.5–13 μm at a 30 Hz image frequency. The emissivity was set to 0.98. ThermaCAM Researcher Pro 2.10 software was later used to view and analyze the temperature readings. During each thermographic session, the temperature and humidity in the milking room was also recorded to adjust the mentioned software.

The glandular thermographic images were analyzed by dividing each gland into five circles (AR01; AR02; AR03; AR04; AR05) in a diagonal line from the nipple to the upper area of the mammary gland, avoiding areas in contact with the inner side of the hind limbs, the medial suspensory ligament, and the groin. The maximum and average temperatures were recorded for each circle, covering about 3674 pixels per circle, which represented 12% of the overall image and approximately 75% of the targeted glandular image. Maximum temperatures were also recorded for the left lacrimal area and perianal regions (Figure 1).

### 2.3. Mammary Glands’ Health Status

The health status of the mammary glands was assessed using microbiological cultures and somatic cell count (SCC) analysis on milk samples collected before milking once a month. For the SCC analysis, the mammary glands were cleaned to remove any gross dirt, if present, and foremilk was discarded. Later, 50 mL samples were manually drawn from each gland prior to machine milking and preserved with azidiol. SCC was quantified as × 10^3^ cells/mL and analyzed at the interprofessional dairy laboratory of Castilla-La Mancha (LILCAM, Castilla-La Mancha, Spain) using a fluoro-opto-electronic method (Fossomatic 5000, Foss, Hillerød, Denmark). Glands with more than 400,000 cells/mL were classified as High SCC, and those with less than 400,000 cells/mL were classified as Low SCC according to the usual procedure in this breed [29]. For the microbiological cultures, following the SCC milk sampling, foremilk was discarded, and the teats were sanitized with 70% alcohol before collecting 5 mL milk samples from each gland for bacteriological testing. These samples were maintained at 4 °C and not kept for longer than 4 h before undergoing bacteriological examination. This examination involved the inoculation of 20 µL of milk onto blood agar plates containing 5% sheep blood (Biomerieux, Lyon, France) and incubating them aerobically at 37 °C. Evaluations were conducted at 48 h. A positive result was classified as the presence of five or more identical colonies, indicating significant bacterial growth. Samples with less than five colonies were classified as negative.

### 2.4. Statistical Analysis

The PROC CORR procedure in SAS (version 9.4) was used to examine the relationships among 15 variables. These variables include a series of measurements related to five areas (AR01 through AR05) on the mammary gland. The maximum (max) and average (avg) values for each of these areas were analyzed. Additionally, the ambient temperature in the milking parlor (AT), rectal temperature (RT), and lacrimal temperature (LT), as well as the somatic cell count (SCC) were included in the analysis.

To know the values of SCC along the lactation and to assess the impact of SCC classification (High SCC and Low SCC) on the studied variables, a linear mixed model (Proc GLIMMIX, SAS, version 9.4) was utilized. In addition to the SCC classification, the fixed effects included in the model were the control (1 to 6) and the AT. To account for repeated measurements taken from the same sheep, each sheep was considered a random effect, and a compound symmetry covariance structure was implemented. Similarly, to determine the influence of microbial growth outcomes on milk samples from different glands, a linear mixed model (Proc GLIMMIX, SAS, version 9.4) was utilized. The analysis categorized the samples into two groups based on the presence of microbial growth, identified as positive (five or more identical colonies) and negative (less than five colonies). In addition to the categorization of microbial growth, the fixed effects included in the model were the control and the AT. Given the repeated sampling from the same sheep, each sheep was considered as a random effect, and a compound symmetry covariance structure was employed to handle the intra-gland correlations.

## 3. Results

Table 1 shows the average values of the variables studied throughout the experiment.

Table 2 presents the correlation matrix for the ambient temperature, the different surface temperature measurements, rectal temperatures, and somatic cell counts obtained. The ambient temperature (AT) displayed high positive correlations with most thermographic measurements (*p* < 0.01), indicating its influence on surface temperature readings. Thus, the values of the surface body temperature increased as lactation progressed (January to July). However, AT showed no correlation with rectal temperature (RT). As expected, high significant positive correlations were also observed among maximum and average temperatures recorded using IRT, including AR01Max, AR02Max, AR03Max, AR04Max, and AR05Max (*p* < 0.01). The average temperatures (AR01Avg to AR05Avg) similarly exhibited high correlations with both each other and the corresponding maximum temperatures, suggesting consistent patterns across the different body regions that were measured.

LT demonstrated significant correlations with most of the maximum and average temperatures recorded via IRT (*p* < 0.01). However, LT did not correlate significantly with RT. PT also showed significant correlations with several thermographic readings (*p* < 0.01) but failed to demonstrate a significant relationship with RT. Moreover, RT exhibited no associations with the other variables in the matrix, showing non-significant correlations with both body surface temperatures (Table 2).

The relationship between the udder health and body temperatures generally revealed low correlations. Specifically, the surface temperatures of the udder showed low negative correlations with AR04Max, AR05Max, AR01Avg, AR02Avg, AR03Avg, AR04Avg, and AR05Avg. Conversely, no correlations were observed with the body surface temperatures of other areas such as PT and LT. Additionally, there was no correlation between the values of the SCC and RT (Table 2).

In the analysis conducted to determine the effect of microbiological growth on surface temperatures, no statistically significant differences were observed across the anatomical regions measured (AR01Max to AR05Max and AR01Avg to AR04Avg). Only the AR5Avg from glands with positive microbiological growth showed a decrease in its value with a difference of 0.9 °C (*p* = 0.04, Table 3).

As lactation progressed, there was a general increase in SCC, rising from 165,000 cells/mL in January to 463,000 cells/mL in July. In the analysis conducted to determine the effects of SCC levels on IRT records, differences were observed across all the measured areas of the mammary glands, both for maximum (Max) and average (Avg) temperatures. Glands with an SCC ≤ 400,000 cells/mL consistently exhibited higher temperatures compared to those with an SCC > 400,000 cells/mL. Specifically, for maximum temperatures, the differences were 1.14 °C for AR01Max, 1.14 °C for AR02Max, 1.16 °C for AR03Max, 1.14 °C for AR04Max, and 1.01 °C for AR05Max (*p* < 0.05). Similarly, for average temperatures, the differences were 1.14 °C for AR01Avg, 1.14 °C for AR02Avg, 1.16 °C for AR03Avg, 1.14 °C for AR04Avg, and 1.01 °C for AR05Avg (*p* < 0.05). These differences were consistent across various regions of the mammary gland and did not vary significantly between maximum and average temperature measurements. Finally, the values of RT and LT were similar across both groups and animals (High and Low SCC). Notably, animals with High SCC exhibited higher PT values than those with Low SCC, although the differences were modest, amounting to only 0.34 °C (Table 3).

## 4. Discussion

The correct IRT technique is crucial in veterinary diagnostics, especially when interpreting thermal images. Factors such as ambient humidity, wind speed, and direct solar exposure can influence these interpretations, potentially affecting the reliability of measurements [30,31]. In covered, well-constructed barns, such as those commonly used for Manchega dairy sheep, these factors should not limit the reliability of data when using IRT in such situations. The relationship between ambient temperature and skin surface temperature in animals is complex, influenced by various environmental and physiological factors. Several studies have demonstrated a correlation between these two temperatures, which can significantly vary depending on the body region measured and the prevailing environmental conditions [5,32,33]. Notably, responses such as cutaneous vasodilation and an increased respiratory rate commonly occur with an increase in ambient temperature, potentially raising skin surface temperatures without significant changes in core body temperature (rectal temperature) [5,34]. Additionally, specific studies on stressed sheep and cows have reported a positive correlation between skin surface temperature and rectal temperature [35,36,37]. However, the correlation between skin surface temperature and rectal temperature tends to decrease in body regions covered with dense fur, wool, or less vascularized skin [5]. In this study, body surface temperature measurements in Manchega sheep varied according to environmental temperature, while rectal temperature remained stable over time. This pattern suggests that Manchega sheep can adapt their physiological responses to different climatic scenarios, maintaining thermal homeostasis even when surface temperatures fluctuate. The stability of rectal temperature under varying environmental conditions has been similarly observed in other studies, such as those conducted by Ibáñez et al. [5], where animals demonstrated the ability to regulate their core body temperature despite external heat exposure. However, it is important to note that this adaptive mechanism might be challenged under extreme climatic conditions, like in extensive systems [38] or during transportation [39], which were not evaluated in the present study.

The literature highlights the capability of IRT to detect stress or diseases marked by an elevation in skin temperature, proving to be useful in pigs, horses, and ruminants [13,40,41,42]. Traditionally, inflammation has been associated with local temperature increases, as evidenced in various studies [10,11,12]. However, the relationship between inflammation and temperature changes is complex and can vary significantly, influenced by several factors. Acute inflammation usually results in a rise in local or systemic temperature due to increased blood flow and cytokine activity [43,44], whereas chronic inflammation may lead to a decrease in temperature due to impaired blood flow or metabolic functions in the affected tissues [15]. This last dynamic has also been observed in sows with chronic puerperal mastitis [45]. Furthermore, the specific location of inflammation plays a critical role as deep tissue inflammation may not produce noticeable changes in surface temperature, while superficial inflammation typically leads to a detectable increase. In the study by Bortolami et al. [15], *S. aureus* and coagulase-negative *staphylococci*, which are primary agents of mastitis in dairy sheep [18,19,20], were identified as causing a decrease in udder surface temperature. Our study suggests that elevated SCC, indicative of inflammation, correlates with a decrease in mammary gland surface temperature. This pattern implies a generalized physiological response where glands with higher SCC levels represent a chronic infection stage, resulting in a decreased surface temperature despite increased cellular activity. These findings contrast with those of other researchers [14,24], possibly due to differences in study design and the specific pathogens investigated, which might mimic an acute infection response. Notably, these employed LPS from *E. coli* to simulate this condition, potentially influencing the distinct thermal responses observed.

The relationship between microbiological milk culture results and SCC did not exhibit concordance in this study as only one variable of the ten studied variables of the udder surface temperature showed differences between animals with or without positive microbiological growth. This may be due to varying immune responses depending on the infecting microorganism or the stage of infection. One possible explanation is that certain pathogens might trigger an immune response robust enough to inhibit bacterial growth, thus preventing detection through standard culture methods despite the presence of infection. Another possibility is the intermittent excretion of microorganisms in milk or their sequestration in specific areas of the mammary gland, which has been demonstrated in other studies [46,47,48,49]. Additionally, the presence of natural inhibitors in milk or limitations of classical culture media could account for the absence of bacterial growth in some cases with an elevated SCC [50].

When working with IRT, especially in the context of monitoring mammary gland inflammation, it is crucial to consider numerous factors that can significantly influence its effectiveness and accuracy. The results found in the literature do not reach a consensus on the utility of IRT for monitoring mammary gland inflammation due to the diverse temperature variations that can occur during an episode of inflammation [10,11,12,15,24,25]. These variations can depend significantly on whether the inflammation is in an initial or advanced stage, whether the animal exhibits additional symptoms, the presence of edema, and whether the inflammation is acute or chronic. This complexity underscores the challenge in using IRT as a definitive tool for diagnosing and managing udder health, suggesting a need for a more nuanced approach that incorporates a range of diagnostic indicators alongside thermographic data. Further research is needed to explore these relationships under varying infection types and environmental conditions.

## 5. Conclusions

IRT has demonstrated that Manchega dairy sheep can adapt to a wide range of temperatures, as shown in this study. While body surface temperatures varied according to ambient conditions, rectal temperature remained constant, maintaining the physiological equilibrium essential for proper metabolic functioning. Additionally, significant correlations were observed among different thermographic measurements across various body regions, indicating consistent temperature patterns. However, SCC exhibited low correlations with both rectal and surface temperatures. The presence of pathogens, as indicated by microbiological cultures, did not significantly alter surface temperatures. Importantly, glands with higher SCC demonstrated reduced temperatures. These findings underscore the complex relationship between temperature measurements and udder health, highlighting the need for integrated diagnostic approaches in sheep health management.

## Figures and Tables

**Figure 1 animals-15-00773-f001:**
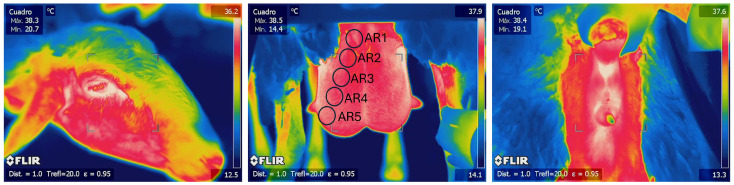
Thermographic images from the lacrimal area, mammary glands (AR01; AR02; AR03; AR04; AR05), and perianal area.

**Table 1 animals-15-00773-t001:** Summary of temperature measurements in Manchega sheep and somatic cell counts throughout the study.

Variable	Average	SD	Minimum	Maximum
AT (°C)	18.70	6.65	9.3	30.0
AR01Max (°C)	35.16	2.52	20.9	39.4
AR02Max (°C)	35.24	2.44	21.8	39.2
AR03Max (°C)	35.23	2.32	21.6	39.4
AR04Max (°C)	34.94	2.26	22.5	38.8
AR05Max (°C)	34.45	2.30	23.2	39.0
AR01Avg (°C)	34.12	2.65	20.2	38.6
AR02Avg (°C)	34.07	2.58	21.1	38.5
AR03Avg (°C)	33.83	2.61	20.8	38.5
AR04Avg (°C)	33.39	2.68	22.5	38.0
AR05Avg (°C)	32.81	2.81	25.1	37.8
PT (°C)	38.69	1.13	34.2	40.9
RT (°C)	38.86	0.361	37.2	39.9
LT (°C)	37.93	0.89	35.1	40.1
SCC (°C)	338	1964	4	27,007

AT: ambient temperature; AR01Max: maximum temperature in AR01; AR02Max: maximum temperature in AR02; AR03Max: maximum temperature in AR03; AR04Max: maximum temperature in AR04; AR05Max: maximum temperature in AR05; AR01Avg: average temperature in AR01; AR02Avg: average temperature in AR02; AR03Avg: average temperature in AR03; AR04Avg: average temperature in AR04; AR05Avg: average temperature in AR05; PT: perianal temperature; RT: rectal temperature; LT: lacrimal temperature; SCC: somatic cell count; SD: standard deviation.

**Table 2 animals-15-00773-t002:** Correlation matrix between measured parameters. The upper triangle in gray corresponds to the significance level (** = *p*-values < 0.01; * = *p*-values < 0.05). The lower triangle corresponds to the values of the correlations.

	AT	AR01 Max	AR02 Max	AR03 Max	AR04 Max	AR05 Max	AR01 Avg	AR02 Avg	AR03 Avg	AR04 Avg	AR05 Avg	PT	RT	LT	SCC
AT	1.00	**	**	**	**	**	**	**	**	**	**	**	0.06	**	0.12
AR01Max	0.72	1.00	**	**	**	**	**	**	**	**	**	**	*	**	0.09
AR02Max	0.76	0.97	1.00	**	**	**	**	**	**	**	**	**	0.09	**	0.10
AR03Max	0.80	0.92	0.95	1.00	**	**	**	**	**	**	**	**	0.44	**	0.07
AR04Max	0.79	0.83	0.87	0.94	1.00	**	**	**	**	**	**	**	0.43	**	*
AR05Max	0.77	0.76	0.80	0.85	0.91	1.00	**	**	**	**	**	**	0.23	**	*
AR01Avg	0.77	0.98	0.96	0.92	0.84	0.79	1.00	**	**	**	**	**	*	**	*
AR02Avg	0.80	0.95	0.98	0.95	0.88	0.81	0.97	1.00	**	**	**	**	0.25	**	*
AR03Avg	0.83	0.90	0.94	0.97	0.93	0.85	0.93	0.97	1.00	**	**	**	0.61	**	*
AR04Avg	0.82	0.82	0.86	0.91	0.96	0.92	0.86	0.89	0.95	1.00	**	**	0.75	**	*
AR05Avg	0.81	0.76	0.79	0.83	0.89	0.96	0.80	0.83	0.87	0.93	1.00	**	0.43	**	*
PT	0.59	0.57	0.57	0.57	0.55	0.52	0.59	0.59	0.59	0.57	0.56	1.00	*	**	0.05
RT	−0.06	0.08	0.05	0.02	0.04	0.07	0.04	0.01	0.02	0.01	0.03	0.19	1.00	0.45	0.06
LT	0.77	0.63	0.64	0.67	0.66	0.65	0.64	0.65	0.68	0.68	0.66	0.58	0.04	1.00	0.79
SCC	−0.05	−0.06	−0.06	−0.06	−0.08	−0.07	−0.07	−0.07	−0.08	−0.08	−0.08	−0.06	−0.06	0.01	1.00

AT: ambient temperature; AR01Max: maximum temperature in AR01; AR02Max: maximum temperature in AR02; AR03Max: maximum temperature in AR03; AR04Max: maximum temperature in AR04; AR05Max: maximum temperature in AR05; AR01Avg: average temperature in AR01; AR02Avg: average temperature in AR02; AR03Avg: average temperature in AR03; AR04Avg: average temperature in AR04; AR05Avg: average temperature in AR05; PT: perianal temperature; RT: rectal temperature; LT: lacrimal temperature; SCC: somatic cell count.

**Table 3 animals-15-00773-t003:** Comparative analysis of body temperatures in Manchega sheep by different levels of SCC and microbiological cultures of the milk (mean ± standard error).

Variable	MG SCC Status	Mean	SR	*p*-Value	Variable	MG MC Status	Mean	SR	*p*-Value
AR01Max (°C)	Low SCC	35.22	0.12	<0.01	AR01Max (°C)	−	35.19	0.12	0.39
	High SCC	34.08	0.36			+	34.97	0.24	
AR02Max (°C)	Low SCC	35.30	0.10	<0.01	AR02Max (°C)	−	35.26	0.11	0.56
	High SCC	34.16	0.34			+	35.12	0.23	
AR03Max (°C)	Low SCC	35.30	0.08	<0.01	AR03Max (°C)	−	35.26	0.09	0.46
	High SCC	34.14	0.31			+	35.10	0.21	
AR04Max (°C)	Low SCC	35.01	0.08	<0.01	AR04Max (°C)	−	34.99	0.09	0.17
	High SCC	33.87	0.31			+	34.68	0.21	
AR05Max (°C)	Low SCC	34.52	0.09	<0.01	AR05Max (°C)	−	34.51	0.09	0.11
	High SCC	33.51	0.31			+	34.14	0.21	
AR01Avg (°C)	Low SCC	34.18	0.12	<0.01	AR01Avg (°C)	−	34.16	0.12	0.27
	High SCC	32.98	0.37			+	33.87	0.25	
AR02Avg (°C)	Low SCC	34.14	0.10	<0.01	AR02Avg (°C)	−	34.12	0.11	0.20
	High SCC	32.95	0.36			+	33.80	0.24	
AR03Avg (°C)	Low SCC	33.89	0.09	<0.01	AR03Avg (°C)	−	33.89	0.10	0.11
	High SCC	32.66	0.35			+	33.48	0.23	
AR04Avg (°C)	Low SCC	33.45	0.10	<0.01	AR04Avg (°C)	−	33.45	0.10	0.08
	High SCC	32.34	0.36			+	32.96	0.24	
AR05Avg (°C)	Low SCC	32.86	0.10	0.02	AR05Avg (°C)	−	32.86	0.11	0.04
	High SCC	31.96	0.38			+	31.96	0.38	
PT (°C)	Low SCC	38.69	0.05	0.52	PT (°C)	−	38.74	0.05	<0.01
	High SCC	38.59	0.16			+	38.40	0.11	
RT (°C)	Low SCC	38.87	0.03	0.12	RT (°C)	−	38.87	0.03	0.80
	High SCC	38.79	0.05			+	38.86	0.04	
LT (°C)	Low SCC	38.01	0.04	0.62	LT (°C)	−	37.94	0.05	0.49
	High SCC	37.92	0.17			+	37.86	0.12	

MG: mammary gland; MC: microbiological cultures; High SCC: >400,000 cels/mL; Low SCC: <400,000 cels/mL; AR01Max: maximum temperature in AR01; AR02Max: maximum temperature in AR02; AR03Max: maximum temperature in AR03; AR04Max: maximum temperature in AR04; AR05Max: maximum temperature in AR05; AR01Avg: average temperature in AR01; AR02Avg: average temperature in AR02; AR03Avg: average temperature in AR03; AR04Avg: average temperature in AR04; AR05Avg: average temperature in AR05; PT: perianal temperature; RT: rectal temperature; LT: lacrimal temperature; SCC: somatic cell count.

## Data Availability

Dataset available on request from the authors.

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
