# Peer review of "Evaluation of Body and Udder Temperatures and Mammary Gland Health Status Throughout Lactation in Manchega Dairy Sheep"

_animals, 2025, doi:10.3390/ani15060773_

Round 1

Reviewer 1 Report

Comments and Suggestions for Authors

Bueso-Ródenas and his team present for review the research work “Evolution of Body and Udder Temperatures and Mammary Health Status Throughout Lactation in Manchega Dairy Sheep”. The authors established the objectives of relating different ways of measuring the body temperature of sheep, subsequently trying to verify whether these were related to changes in the ubber’s health of animals affected with high SCC counts. Firstly, we would like to congratulate the authors for the initiative and the results obtained. However, several conclusions that were proposed to be obtained did not have clear results, leaving doubts about the relationships between the variables. However, these are also good results, as the authors were able to present and justify the complexity of the topic in the discussion.

Generally, we consider the text to be well written, requiring only a moderate to small revision of the English language. Specifically, we comment on each chapter and present some suggestions that authors may want to consider.

. Abstract – Simple and direct in the objectives to be achieved, also presenting the main conclusions.

. Introduction – Generally well written, it simply addresses the topics involved in the study. We believe that a small paragraph describing the Manchega sheep should be included. Some data about the breed is important to understand the study: indigenous or not? Reference to some previous studies on the breed on the microbial population that affects it most frequently in terms of the cause of mastitis, …

. Regarding the study design presented in the “Material and Methods” chapter, it is well outlined. It should be noted that we consider the number of animals used in the study to be quite adequate, making it possible to obtain robust data.

- Little magpie: They refer to image 1 in the text and present figure 1 in the caption.

. Results – clear presentation. Tables allow you to easily query data.

. Discussion – the results obtained were well explored and justified.

. Limited variety of bibliographic references.

Comments on the Quality of English Language

Overall, its well written. Moderate to minor English revision should be done.

Line 15 – delete “the”… suggesting sheep can…

Line 47 – delete “the” - … between rectal and with the temperature…

Line 47 – define IRT or in Line 41 (add IRT after infrared thermography)

Line 48 – in the other hand… to use this there must be “in one hand”… if going with “In this line of research…” and then “on the other point of view of this research…” maybe? If using this sentence, its “on the other hand” and not “in”.

Line 48 – the was low? Or “no correlations” – plural

Line 49 – or the sheep? Or do you meant “surfaces of the sheep” ?

Line 60 – consulted literature – would be better to add references… which literature? I can see then that you have all the references – so maybe the sentence is not appropriate? Articles should be easy to read and with no repeating information.

Line 66 – on the other hand, same question, what’s the other hand/point of view?

Line 70 – on the contrary? Understand your way of writing to show opposite and different ideas, but rephrase this sentence please.

Line 86 – intramammary? Or just mammary?

Line 102 – patios? Yards?

Line 103 – delete “months”…”between January and July.

Line 127 – in the rectum. Full stop mark is missing here

Line 167/184 – with less … instead of fewer…

Line 227 – relationship singular

Line 239 – significant differences? Are they statistically significant? Answer on line 244 – rewrite this paragraph please..

Line 262 – delete “the”… “correct ultrasound…”

Line 274 – pay attention to “on the other hand” throughout the text…

Line 277 – temperature isn’t strong neither weak, its low or high…

Line 289 – not necessary to repeat what IRT is… it should be explained on line 47

Line 290 – pathologies? Or diseases?

Line 290 - … proving to be useful …

Line 300 – observable? Or evident? Not all studies used IRT, they had to measure manually the temperature, am I right?

Author Response

REVIEWER

Bueso-Ródenas and his team present for review the research work “Evolution of Body and Udder Temperatures and Mammary Health Status Throughout Lactation in Manchega Dairy Sheep”. The authors established the objectives of relating different ways of measuring the body temperature of sheep, subsequently trying to verify whether these were related to changes in the ubber’s health of animals affected with high SCC counts. Firstly, we would like to congratulate the authors for the initiative and the results obtained. However, several conclusions that were proposed to be obtained did not have clear results, leaving doubts about the relationships between the variables. However, these are also good results, as the authors were able to present and justify the complexity of the topic in the discussion.

Generally, we consider the text to be well written, requiring only a moderate to small revision of the English language. Specifically, we comment on each chapter and present some suggestions that authors may want to consider.

AUTHORS

Thank you for your thoughtful review and the constructive feedback on our manuscript. We are grateful for your compliments on the initiative and the results obtained, and we appreciate your recognition of the complexities involved in our study.

We are committed to addressing the concerns you raised regarding the clarity of certain conclusions and the relationships between variables. Your insightful suggestions for revising the manuscript are all well-received and will be incorporated into our revised submission.

Additionally, we will undertake the moderate language revisions suggested to improve the clarity and quality of our text.

REVIEWER

Abstract – Simple and direct in the objectives to be achieved, also presenting the main conclusions.

AUTHORS

Thank you for your observation.

REVIEWER

Introduction – Generally well written, it simply addresses the topics involved in the study. We believe that a small paragraph describing the Manchega sheep should be included. Some data about the breed is important to understand the study: indigenous or not? Reference to some previous studies on the breed on the microbial population that affects it most frequently in terms of the cause of mastitis.

AUTHORS

Thank you for your observation. We agree with the reviewer and have added this information

IRT has also revealed that well-adapted indigenous breeds, such as the Manchega sheep, possess the ability to dissipate excess body heat under hot conditions through increased respiration rates and vasodilation among other adaptive capabilities. [5].

In small ruminants, such as Manchega sheep, mastitis is commonly caused by infectious bacteria, including various species of Staphylococcus, which often lead to a high prevalence of subclinical infections [18, 19, 20].

  1. De la Cruz, M.; Serrano, E.; Montoro, V.; Marco, J.; Romeo, M.; Baselga, R.; Albizu, I.; Amorena, B. Etiology and Prevalence of Subclinical Mastitis in the Manchega Sheep at Mid-Late Lactation. Small Rumin. Res. 1994, 14(2), 175–180.
  2. Ariznabarreta, A.; Gonzalo, C.; San Primitivo, F. Microbiological Quality and Somatic Cell Count of Ewe Milk with Special Reference to Staphylococci. J. Dairy Sci. 2002, 85(6), 1370–1375.
  3. Gelasakis, A.I.; Mavrogianni, V.S.; Petridis, I.G.; Vasileiou, N.G.C.; Fthenakis, G.C. Mastitis in Sheep – The Last 10 Years and the Future of Research. Vet. Microbiol. 2015, 181(1–2), 136–146.

REVIEWER

Regarding the study design presented in the “Material and Methods” chapter, it is well outlined. It should be noted that we consider the number of animals used in the study to be quite adequate, making it possible to obtain robust data.

- Little magpie: They refer to image 1 in the text and present figure 1 in the caption.

AUTHORS

Sorry for the mistake.

REVIEWER

. Results – clear presentation. Tables allow you to easily query data.

. Discussion – the results obtained were well explored and justified.

. Limited variety of bibliographic references.

AUTHORS

We have added three references according to the suggestion of the reviewer.

REVIEWER

Comments on the Quality of English Language

Overall, its well written. Moderate to minor English revision should be done.

Line 15 – delete “the”… suggesting sheep can…

AUTHORS

Thank you for your observation.

REVIEWER

Line 47 – define IRT or in Line 41 (add IRT after infrared thermography)

AUTHORS

Thank you for your observation. We have unified in the entire text to IRT

REVIEWER

Line 48 – in the other hand… to use this there must be “in one hand”… if going with “In this line of research…” and then “on the other point of view of this research…” maybe? If using this sentence, its “on the other hand” and not “in”.

AUTHORS

Thank you for your observation. We have changed the construction “in (on) the other hand” by other constructions (other studies, finally, however, additionally…).

REVIEWER

Line 48 – the was low? Or “no correlations” – plural

AUTHORS

According to the suggestion of the other reviewer we have added (and corrected the English according to your suggestion) this part to the discussion section.

REVIEWER

Line 49 – or the sheep? Or do you meant “surfaces of the sheep” ?

AUTHORS

According to the suggestion of the other reviewer we have added (and corrected the English according to your suggestion) this part to the discussion section.

REVIEWER

Line 60 – consulted literature – would be better to add references… which literature? I can see then that you have all the references – so maybe the sentence is not appropriate? Articles should be easy to read and with no repeating information.

AUTHORS

We have included references in the introductory sentence of this paragraph to avoid misunderstandings. We have simplified the text to avoid repeating information. Our goal is to clarify that there is no consensus in the literature, and then to present the main conclusions of various articles.

REVIEWER

Line 66 – on the other hand, same question, what’s the other hand/point of view?

AUTHORS

We have clarified this point in previous observations of the reviewer.

REVIEWER

Line 70 – on the contrary? Understand your way of writing to show opposite and different ideas but rephrase this sentence please.

AUTHORS

We have used the expression “however” to clarify.

REVIEWER

Line 86 – intramammary? Or just mammary?

AUTHORS

The term 'intramammary infection' is commonly used in the literature of this scientific area. However, we have opted for 'mammary infection', as suggested by the reviewer as in this context, as we believe both terms are applicable.

REVIEWER

Line 102 – patios? Yards?

AUTHORS

Yards is more appropriate, thanks for your help

REVIEWER

Line 103 – delete “months”…”between January and July.

AUTHORS

Thanks for your help

REVIEWER

Line 127 – in the rectum. Full stop mark is missing here

AUTHORS

We have completed the sentence

A digital thermometer Vet 12 (Hauptner and Herberholz, Solingen, Germany) was in-serted 3 cm in the rectum to record rectal temperature.

REVIEWER

Line 167/184 – with less … instead of fewer…

AUTHORS

Thanks for your help

REVIEWER

Line 227 – relationship singular

AUTHORS

Thanks for your help

REVIEWER

Line 239 – significant differences? Are they statistically significant? Answer on line 244 – rewrite this paragraph please..

AUTHORS

Sorry for the mistake, we have changed this part of the text:

“In the analysis conducted to determine the effect of somatic cell count (SCC) levels on IRT records, differences were observed across all measured areas of the mammary glands, both for maximum (Max) and average (Avg) temperatures”

REVIEWER

Line 262 – delete “the”… “correct ultrasound…”

AUTHORS

Sorry for the mistake

REVIEWER

Line 274 – pay attention to “on the other hand” throughout the text…

AUTHORS

We have clarified this point in previous observations of the reviewer

REVIEWER

Line 277 – temperature isn’t strong neither weak, its low or high…

AUTHORS

Thank you for your help. We have revised the entire manuscript and used low, decrease, high or increase instead of other terms as weak, weaken or strong.

REVIEWER

Line 289 – not necessary to repeat what IRT is… it should be explained on line 47

AUTHORS

Thank you for your observations

REVIEWER

Line 290 – pathologies? Or diseases?

AUTHORS

We have changed the text according to your suggestion

REVIEWER

Line 290 - … proving to be useful …

AUTHORS

We have changed the text according to your suggestion

REVIEWER

Line 300 – observable? Or evident? Not all studies used IRT, they had to measure manually the temperature, am I right?

AUTHORS

We have changed the text: “detectable”

Reviewer 2 Report

Comments and Suggestions for Authors

Thank for the submitted paper. I have a few comments on it.
- The Title. Is "evolution" a proper word here, shouldn't it be "evaluation"? If it is "evolution", why there are no changes shown in the text? Also add "gland", after the word mammary;
- Some part of the Introduction sounds more like part of discussion (e.x. line 46- 53), rewrite this chapter;
- Correct the use of abbrevietions, add (IRT) in line 42, explain LPS (line 62) and USST (line 71); also there is no need to explain the abbreviations through the entire text, it should be explained the first time it appears in the text;
- Unify the way of writting E. coli, it is a latin name, so write it in italics, also check other latin names in the text;
- The Materials  and Methods. Shouldn't the milk for the SCC analysis be collected the same way as collected for the microbiological cultures? Did you clean the udder before collecting milk for the SCC analysis and collect it after the foremilk?
- Why the changes in temperatures and SCC throughout the whole experiment hasn't been showed? Especially because there is mention about it in the Discussion, and also there may be changes during lactation in this parameter (SCC).
- line 145, is it "Image 1" or "Figure 1"?
- There are two subchapters 2.3 (lines 150 and 168);
- Table 1. and 3., it is good to add informations about units, because e.x. temperature can be in different units;
- Table 3. there are no informations what MG and MC are;
- Why it is mentioned here about "ultrasound technique"?

Author Response

REVIEWER

Thanks for the submitted paper. I have a few comments on it.

AU: Many thanks for your help to improve the quality of our paper. We have taken into account all of your insightful comments.

REVIEWER

The Title. Is "evolution" a proper word here, shouldn't it be "evaluation"? If it is "evolution", why are there no changes shown in the text? Also add "gland", after the word mammary.

AU: Many thanks for your help, we have included these changes which improve the clarity of the title.

REVIEWER

Some part of the Introduction sounds more like part of discussion (e.x. line 46- 53), rewrite this chapter.

AU: Thanks for your comment. We have simplified this part of the introduction as it is also mentioned in the discussion section

REVIEWER

Correct the use of abbreviations, add (IRT) in line 42, explain LPS (line 62) and USST (line 71); also there is no need to explain the abbreviations through the entire text, it should be explained the first time it appears in the text;

AU: We have corrected these abbreviations, and they are only explained the first time they appear in the text.

REVIEWER

Unify the way of writing E. coli, it is a latin name, so write it in italics, also check other latin names in the text.

AU: Thank you for your observation.

REVIEWER

The Materials and Methods. Shouldn't the milk for the SCC analysis be collected the same way as collected for microbiological cultures? Did you clean the udder before collecting milk for the SCC analysis and collect it after the foremilk?

AU: We have changed this part of this section to clarify: For the SCC analysis, the mammary glands were cleaned to remove any gross dirt, if present, and the foremilk was discarded.

REVIEWER

Why the changes in temperatures and SCC throughout the whole experiment hasn't been showed? Especially because there is mention about it in the Discussion, and also there may be changes during lactation in this parameter (SCC).

AU: We fully agree with the reviewer and added this important lacking information.

Thus, the values of the surface body temperature increased as lactation progressed (January to July).

As lactation progressed, there was a general increase in SCC, rising from 165,000 cells/mL in January to 463,000 cells/mL in July.

We have also added the information in the Statistic section

To know the values of SCC along the lactation…

REVIEWER

line 145, is it "Image 1" or "Figure 1"?

AU: Sorry for the mistake

REVIEWER

There are two subchapters 2.3 (lines 150 and 168);

AU: Thanks for your observation

REVIEWER

- Table 1. and 3., it is good to add information about units, because e.x. temperature can be in different units;

AU: We agree with the reviewer and added this information in table 1 and 3 to express the units employed (values of temperatures in °C)

REVIEWER

Table 3. there are no information what MG and MC are;

AU: We have added this important lacking information in Table 3

REVIEWER

Why is it mentioned here about "ultrasound technique"?

AU: Sorry for the mistake, it is IRT, of course.

Round 2

Reviewer 2 Report

Comments and Suggestions for Authors

Thank you for all the amendments made in the article.

I have just two small comments on the revised article. I do not think that the "somatic cells count" needs to be capitalized in the text. I also think that in tables there is no need to add (°C) in all rows. You could also mention in the table title that the temperature is in °C.